# Fucoxanthin Attenuates the Reprogramming of Energy Metabolism during the Activation of Hepatic Stellate Cells

**DOI:** 10.3390/nu14091902

**Published:** 2022-05-01

**Authors:** Minkyung Bae, Mi-Bo Kim, Ji-Young Lee

**Affiliations:** 1Department of Nutritional Sciences, University of Connecticut, Storrs, CT 06269-4017, USA; mkbae@changwon.ac.kr (M.B.); mi-bo.kim@uconn.edu (M.-B.K.); 2Department of Food and Nutrition, Interdisciplinary Program in Senior Human Ecology, BK21 FOUR, College of Natural Sciences, Changwon National University, Changwon 51140, Korea

**Keywords:** fucoxanthin, mitochondrial respiration, glycolysis, fibrogenesis, hepatic stellate cell

## Abstract

Hepatic stellate cells (HSC) play a major role in developing liver fibrosis. Upon activation during liver injury, activated HSC (aHSC) increase cell proliferation, fibrogenesis, contractility, chemotaxis, and cytokine release. We previously showed that aHSC have increased mitochondrial respiration but decreased glycolysis compared to quiescent HSC (qHSC). We also demonstrated that fucoxanthin (FCX), a xanthophyll carotenoid, has an anti-fibrogenic effect in HSC. The objective of this study was to investigate whether FCX attenuates metabolic reprogramming occurring during HSC activation. Mouse primary HSC were activated in the presence or absence of FCX for seven days. aHSC displayed significantly decreased glycolysis and increased mitochondrial respiration compared to qHSC, which was ameliorated by FCX present during activation. In addition, FCX partially attenuated the changes in the expression of genes involved in glycolysis and mitochondrial respiration, including hexokinase 1 (*Hk1*), *Hk2*, peroxisome proliferator-activated receptor γ coactivator 1β, and pyruvate dehydrogenase kinase 3. Our data suggest that FCX may prevent HSC activation by modulating the expression of genes crucial for metabolic reprogramming in HSC.

## 1. Introduction

Liver fibrosis results from sustained wound healing in response to the chronic liver injury induced by chronic hepatitis C virus infection, alcohol abuse, non-alcoholic steatohepatitis, metabolic diseases, and parasitic infection [1,2,3]. Liver fibrosis is characterized by abnormal accumulation of extracellular matrix (ECM) proteins, disrupting hepatic architecture and eventually liver functions [2]. Various factors regulate fibrosis progression, including genetic polymorphisms and epigenetic marks [4]. It is important to identify potential anti-fibrotic agents that can prevent or reverse liver fibrosis.

In the fibrotic liver, hepatic stellate cells (HSC) produce ECM proteins, such as fibrillar collagens and basement membrane proteins [5]. HSC are pericytes in the liver residing in the space between hepatocytes and sinusoidal endothelial cells [6]. One of the major physiological functions of HSC is regulating vitamin A homeostasis by storing vitamin A in their intracellular lipid droplets in the normal liver [7]. HSC also play an important role in maintaining hepatic architecture by regulating ECM turnover through the production of ECM proteins, matrix metalloproteinases, and tissue inhibitors of metalloproteinase (TIMPs) [6]. However, excess production by activated HSC (aHSC) and deposition of ECM can lead to liver fibrosis. Therefore, inhibiting dysregulated activation of HSC may be beneficial for preventing liver fibrosis [8].

The major metabolic pathways for ATP generation in cells are glycolysis and mitochondrial respiration [9]. The reprogramming of cells for their preference to produce ATP has been extensively investigated since the observation of tumor cells’ aerobic glycolysis, called the Warburg effect [10]. Furthermore, utilization of glycolysis and mitochondrial respiration for producing energy is critical for the cells’ activation or differentiation and furthers their functions in various cell types, including T cells [11], macrophages [12], and mast cells [13]. A recent study has shown that differentiation of fibroblast into myofibroblast induced by transforming growth factor β1 (TGFβ1), a fibrogenic cytokine, was accompanied by an increase in mitochondrial DNA content and mitochondrial respiration [14]. In addition, we previously demonstrated that aHSC have increased mitochondrial respiration and decreased glycolysis compared with qHSC [15,16]. As the reprogramming of HSC for their energy metabolism is likely essential for their activation and functions, regulating this process may be effective in controlling HSC activation.

Fucoxanthin (FCX) is a xanthophyll carotenoid abundant in brown seaweeds and diatoms [17]. It has been suggested that FCX may be beneficial for the prevention of various chronic diseases, such as cancer, obesity, diabetes mellitus, and liver diseases [18]. We previously demonstrated that FCX exerts anti-fibrogenic effects by attenuating the activation of SMA- and MAD-related protein (SMAD3) and reducing the accumulation of reactive oxygen species (ROS) in HSC [19]. However, the effect of FCX on HSC energy metabolism has not been studied. Therefore, in the present study, we sought to investigate whether FCX alters metabolic reprograming of HSC during their activation.

## 2. Materials and Methods

### 2.1. Primary Mouse HSC Isolation and Culture

Primary mouse HSC were isolated from the liver of C57BL/6J mice using the pronase and collagenase digestion method as described in detail previously [20]. To activate the cells, HSC were cultured on uncoated plastic dishes (BD Falcon, Franklin Lakes, NJ, USA) [21] in low-glucose Dulbecco’s modified Eagle medium supplemented with fetal bovine serum (10%), L-glutamine (4 mM), penicillin (100 U/mL), and streptomycin (100 μg/mL) as previously described [20,22]. The cells were maintained at 37 °C under 5% CO_2_. Cell culture supplies were purchased from HyClone (Thermo Scientific, Logan, UT, USA). The cells at day 1 and day 7 after isolation were considered as qHSC and aHSC, respectively.

### 2.2. FCX Treatment

FCX was purchased from Sigma-Aldrich (St. Louis, MO, USA). The preparation of FCX stock and FCX-containing media was previously described [19]. Primary mouse HSC were treated with FCX (2 or 3 μM) from day 2 to day 7 with daily media change.

### 2.3. Energy Metabolism of Cells

qHSC and aHSC treated with or without FCX (3 μM) during activation were subjected to an XFe24 Extracellular Flux Analyzer (Seahorse Bioscience, North Billerica, MA, USA) for Mito Stress test and Glycolysis Stress test (Seahorse Biosciences) as previously described [16,23]. Once the assay was finished, total DNA was isolated from each well using NucleoSpin^®^ Tissue (MACHEREY-NAGEL Inc, Dueren, Germany) for data normalization.

### 2.4. Quantitative Real-Time PCR (qRT-PCR)

The total RNA was isolated from primary mouse qHSC and aHSC treated with or without FCX (2 or 3 μM) to synthesize cDNA for gene expression analysis using the SYBR green method in a Bio-Rad CFX96 Real-Time System (Bio-Rad, Hercules, CA, USA) as previously described [24,25].

### 2.5. Statistical Analysis

One-way analysis of variance (ANOVA) with Newman–Keuls post hoc analysis was conducted using GraphPad Prism 9.0 (GraphPad Software, La Jolla, CA, USA). *p* values less than 0.05 were considered statistically significant. All values were expressed as mean ± standard error of the mean.

## 3. Results

### 3.1. FCX Abolished the Induction of Collagen Genes during HSC Activation

In our previous study, FCX showed anti-fibrogenic properties evidenced by reduced mRNA levels of fibrogenic genes, such as α smooth muscle actin (*Acta2*) and procollagen type I α1 (*Col1a1*), in primary mouse HSC [19]. To confirm the anti-fibrogenic effect of FCX in HSC, the expression of other collagens, including *Col1a2*, *Col3a1*, *Col6a1*, and *Col6a3*, was measured in primary mouse qHSC and aHSC treated with or without FCX (2 or 3 μM). The expression of *Col1a2*, *Col3a1*, and *Col6a3* was markedly increased in aHSC compared with qHSC (Figure 1). Among them, FCX treatment (3 μM) significantly decreased *Col1a2* and *Col3a1* mRNA abundance in aHSC. As 3 μM of FCX treatment had a stronger effect than 2 μM of FCX on reducing collagen expression, we used 3 μM of FCX in the following experiments to determine its role in metabolic reprogramming during HSC activation.

### 3.2. Decreased Glycolysis in Primary Mouse aHSC was Inhibited by FCX

The rate of glycolysis was measured by the extracellular acidification rate (ECAR) in primary mouse qHSC and aHSC treated with or without FCX (3 μM). aHSC showed significantly lower glycolysis than qHSC, which was abrogated when FCX was added during HSC activation (Figure 2A,B). Glycolytic capacity, i.e., the maximal capacity of glycolysis, and non-glycolytic acidification were significantly increased in aHSC compared with qHSC, and FCX further increased these parameters in aHSC. Glycolytic reserve, indicating the capability to increase glycolysis in response to energy demand, was significantly increased in aHSC compared to qHSC regardless of FCX treatment.

### 3.3. FCX Partially Attenuated Changes in the Expression of Genes Involved in Glycolysis during HSC Activation

As FCX treatment during HSC activation prevented the decrease in glycolysis in aHSC, we further investigated whether FCX altered the expression of genes related to glycolysis. We previously identified that the major glucose transporter in HSC is glucose transporter 1 (GLUT1) [16]. The expression of *Glut1* and its transcription factor hypoxia-inducible factor-1α (*Hif1a*) was significantly decreased in aHSC compared to qHSC, which was further significantly decreased by FCX (Figure 3A). Once glucose is taken up by cells, it is converted into glucose-6-phosphate by hexokinase (HK), one of the rate-limiting steps in glycolysis [26]. The mRNA levels of *Hk1* and *Hk2* were significantly higher in aHSC than in qHSC only in the absence of FCX during HSC activation. Pyruvate, the end product of glycolysis, can be converted into lactate by lactate dehydrogenase [27], and lactate is transported across the cell membrane through monocarboxylate transporters (MCT), or members of the solute carrier family 16, such as solute carrier family 16 member 1 (*Slc16a1*) [28]. There were no significant differences in mRNA levels of lactate dehydrogenase a (*Ldha*) and *Slc16a1* between qHSC and aHSC regardless of FCX (Figure 3B). The pyruvate dehydrogenase (PDH) complex catalyzes the conversion of pyruvate into acetyl-CoA. PDH is active when it is dephosphorylated and its activity is primarily regulated via phosphorylation by pyruvate dehydrogenase kinase (PDK) [29]. The expression of *Pdk1*, *Pdk2*, and *Pdk3* was higher in aHSC than in qHSC (Figure 3C). When the cells were treated with FCX during their activation, the mRNA level of *Pdk1* was further increased; however, that of *Pdk3* was significantly decreased. *Pdk3* was more abundant in HSC than *Pdk1*.

### 3.4. FCX Inhibited an Increase in Mitochondrial Respiration in Primary Mouse aHSC

Mitochondrial respiration was measured by the oxygen consumption rate (OCR) in qHSC, aHSC, and aHSC treated with FCX during activation (Figure 4A). Basal respiration, ATP production, and OCR/ECAR ratio were significantly higher in aHSC than those in qHSC, which were abolished by FCX (Figure 4B). Furthermore, aHSC showed a trend toward an increase in non-mitochondrial respiration, which was significantly different from FCX-treated aHSC. However, basal glycolysis, maximal respiration, spare respiratory capacity, and proton leak were not significantly different between groups.

### 3.5. FCX Prevented a Reduction in Ppargc1a Expression in Primary Mouse aHSC

We further examined the effect of FCX on the expression of genes related to mitochondrial biogenesis and mitochondrial respiration during HSC activation. Mitochondrial transcription factor A (TFAM) is a DNA-binding protein essential for mitochondrial genome replication and transcriptional activation of mitochondrial genes encoding 13 components of the respiratory chain, and therefore it plays a central role in oxidative phosphorylation [30]. Peroxisome proliferator-activated receptor gamma coactivator 1 alpha (PGC-1α) and PGC-1β can induce the expression of mitochondrial genes involved in fatty acid β-oxidation, citric acid cycle, and oxidative phosphorylation [31]. The expression of *Tfam* was decreased by ~50% in aHSC compared to qHSC, which was not altered by FCX treatment (Figure 5A). In addition, *Ppargc1a* mRNA was not significantly changed during HSC activation and by FCX, but *Ppargc1b* mRNA was significantly lower in aHSC than qHSC, but FCX treatment (3 μM) abrogated the decrease (Figure 5B).

## 4. Discussion

HSC play a pivotal role in developing hepatic fibrosis as they are major cells producing ECM proteins in the liver [2]. Activation of HSC in response to hepatic injury requires reprogramming of cells’ energy metabolism to enhance cell proliferation, contractility, fibrogenesis, chemotaxis, and cytokine release [21]. In support of this, our previous studies demonstrated that aHSC has increased mitochondrial respiration but decreased glycolysis compared to qHSC [15,16]. We also previously demonstrated that FCX exerts anti-fibrogenic effects in HSC [19]. However, it was unknown whether FCX alters cellular energy metabolism in HSC. In the present study, we found that FCX inhibited increased mitochondrial respiration and decreased glycolysis during HSC activation. In addition, FCX attenuated the changes in the expression of several genes involved in glucose metabolism and mitochondrial respiration. Our findings suggest that FCX may exert anti-fibrogenic actions by regulating the reprogramming of energy metabolism in HSC.

FCX is the major xanthophyll carotenoid in brown seaweeds [32]. We recently reported that FCX exerts anti-fibrogenic effects in various HSC models, including human HSC line LX-2 cells, primary human HSC, and primary mouse HSC [19]. The anti-fibrogenic effect of FCX was mediated by attenuating the activation of SMAD3, a major component of the TGFβ signaling pathway. In addition, FCX showed an antioxidant capacity as it reduced cellular ROS accumulation [19]. The antioxidant effect of FCX was likely mediated by decreasing the expression of NADPH oxidase 4 (*Nox4*) in LX-2 cells [19]. NOX family includes NOX1, NOX3, NOX4, NOX5, DUOX1, and DUOX2, generating superoxide by transporting electrons across cell membranes [33]. In addition, Takatani et al. [34] reported that FCX administration significantly attenuated liver steatosis in a mouse model of diet-induced non-alcoholic steatohepatitis. FCX not only reduced lipid accumulation in the liver but also decreased hepatic lipid oxidation and expression of inflammatory cytokines, including tumor necrosis factor α, chemokine (C-C motif) ligand 2, interleukin-6 (Il-6), and Il-1β [34]. Moreover, FCX significantly decreased mRNA levels of fibrogenic genes, such as *Tgfb1*, *Col1a1*, *Timp1*, and *Acta2*, in the mouse liver [34]. However, the molecular mechanisms for the anti-fibrogenic effect of FCX in the liver or HSC have been poorly investigated.

Cells generate energy mainly via mitochondrial respiration and glycolysis [9]. Depending on cells’ status and microenvironment, metabolic adaptation occurs to meet their energy demand [35]. One of the most well-characterized metabolic adaptations is aerobic glycolysis, as known as the Warburg effect [10]. Through aerobic glycolysis, cancer cells can rapidly synthesize ATP and support the biosynthetic requirements for their proliferation [36]. Immune cells also undergo metabolic reprogramming upon their activation to exert their appropriate responses to stimuli [37]. Moreover, a recent study showed that TGFβ1-induced differentiation of fibroblasts into myofibroblasts was accompanied by increased mitochondrial DNA content and respiration [14]. In addition, we previously demonstrated that aHSC have higher mitochondrial respiration and lower glycolysis than qHSC [15,16], suggesting that HSC energy metabolism may be critical for their activation and functions. In the present study, FCX completely inhibited those metabolic changes in mitochondrial respiration and glycolysis during HSC activation, indicating the anti-fibrogenic effect of FCX is at least partially mediated by its ability to regulate energy metabolism in HSC.

Glycolysis is a metabolic pathway converting glucose into pyruvate [38]. In the present study, we found that glycolysis was markedly reduced in aHSC compared to qHSC, which was inhibited by FCX treatment. FCX did not alter glycolytic reserve, the amount of glycolysis that cells could increase in response to energy demand [39]. However, FCX significantly increased glycolytic capacity, i.e., the maximal capacity of glycolysis and non-glycolytic acidification in aHSC. We previously identified that GLUT1 is the major glucose transporter in HSC [16]. aHSC had a significantly lower expression of *Glut1* and its transcription factor *Hif1a*, which were further reduced by FCX during HSC activation. Thus, the increase in glycolysis by FCX may not be related to glucose entry in HSC. However, the expression of *Hk1* and *Hk2* showed a trend toward a decrease by FCX, which was significantly increased in aHSC compared to qHSC. Once glucose is taken up by cells through GLUT, it is converted into glucose-6-phosphate by HK, which is necessary for glucose uptake by facilitated diffusion through GLUT. Although glucose-6-phosphate is an inhibitor of HK, studies have shown that the upregulation of HK is related to an increase in glycolysis [40,41]. Cardiac-specific overexpression of HK in mice increased glycolysis and glycogen storage in the heart [40]. In addition, HK2 is known to be upregulated in tumors with increased glycolysis [41]. Therefore, FCX is likely to alter glycolysis by regulating the conversion of glucose into glucose-6-phosphate. Pyruvate, the end product of glycolysis, can be converted into lactate by LDH. Lactate can be transported out of cells by MCT or solute carrier family [42], and *Slc16a1* encodes MCT1 known to be expressed in HSC [43]. The expression of *Ldha* and *Slc16a1* was not changed during HSC activation and by FCX treatment in HSC. Our data suggest that increased glycolysis by FCX is unlikely related to increases in the conversion of pyruvate into lactate or their transport via MCT1. FCX may reduce the conversion of pyruvate into acetyl-CoA, limiting the TCA cycle and mitochondrial respiration in aHSC, as discussed in more detail below.

We found in the present study that FCX inhibited an increase in mitochondrial respiration in aHSC compared to qHSC. aHSC significantly increased basal respiration and ATP production compared to qHSC, which were attenuated by FCX. The effect of FCX on the decrease in mitochondrial respiration is likely mediated, at least in part, by inhibiting the changes in the expression of genes involved in mitochondrial respiration, including *Ppargc1b* and *Pdk3*. PGC-1 family members regulate various cellular processes, including mitochondrial biogenesis, fatty acid β-oxidation, thermogenesis, and gluconeogenesis [44]. PGC1β is known to stimulate mitochondrial respiration [44]. In the present study, when mitochondrial respiration was increased during HSC activation, the expression of *Ppargc1b* was decreased. Therefore, it is presumable that aHSC may undergo mitochondrial stress because they have to process a higher level of mitochondrial respiration with limited mitochondria due to repressed mitochondrial biogenesis. FCX, however, reduced mitochondrial respiration while increased *Ppargc1b* expression in aHSC to similar levels of qHSC. Moreover, pyruvate is converted into acetyl-CoA by the PDH complex in mitochondria, whose activity is decreased by phosphorylation mediated by PDK [45]. In this study, the expression of *Pdk1*, *Pdk2*, and *Pdk3* was increased in aHSC compared to qHSC. Among those genes, FCX significantly attenuated the increase in *Pdk3* expression in aHSC. In our previous study, the increase in the expression of *Pdk2* and *Pdk3* was in parallel with the increased phosphorylation of PDH in primary mouse aHSC compared to qHSC [16]. Our data suggest that FCX decreases mitochondrial respiration by partially altering the expression of genes involved in mitochondrial respiration.

## 5. Conclusions

Our study provides new insight into the molecular mechanisms by which FCX exerts its anti-fibrogenic effect in HSC. The activation of HSC requires metabolic reprogramming in energy-generating pathways. FCX inhibited a decrease in glycolysis and an increase in mitochondrial respiration during HSC activation. The metabolic changes in HSC are needed for their activation and functions; thus, supporting HSC energy metabolism can be a potential target for the prevention of liver fibrosis. Further study is needed to directly evaluate the anti-fibrogenic actions of FCX in vivo using experimental models of liver fibrosis.

## Figures and Tables

**Figure 1 nutrients-14-01902-f001:**
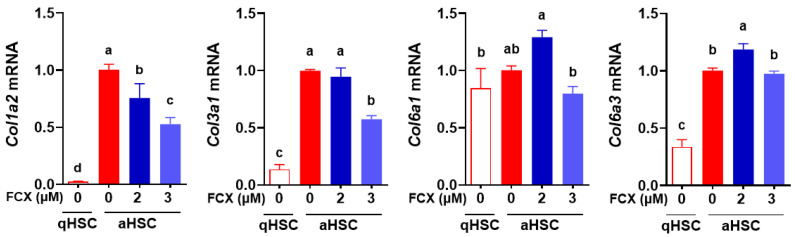
Anti-fibrogenic effects of FCX in HSC. Primary mouse HSC were isolated and cultured for 1 day or 7 days to represent qHSC or aHSC, respectively. FCX (2 or 3 µM) was treated during HSC activation. RT-qPCR analysis was conducted to measure the expression of collagens, including *Col1a2*, *Col3a1*, *Col6a1*, and *Col6a3*. Bars not sharing a common letter are significantly different (*p* < 0.05). Mean ± SEM.

**Figure 2 nutrients-14-01902-f002:**
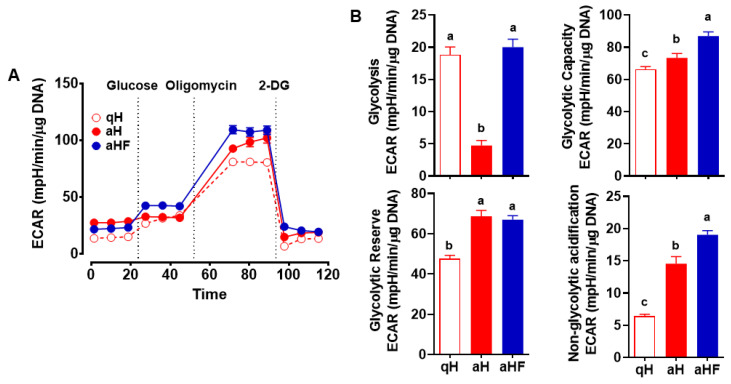
Inhibition of changes in glycolysis by FCX in aHSC. Primary HSC isolated from mice were cultured for 1 day or 7 days in the absence or presence of 3 µM FCX to represent qHSC (qH), aHSC (aH), or FCX-treated aHSC (aHF), respectively. Glycolysis Stress test was conducted (**A**,**B**). Bars not sharing a common letter are significantly different (*p* < 0.05). Mean ± SEM.

**Figure 3 nutrients-14-01902-f003:**
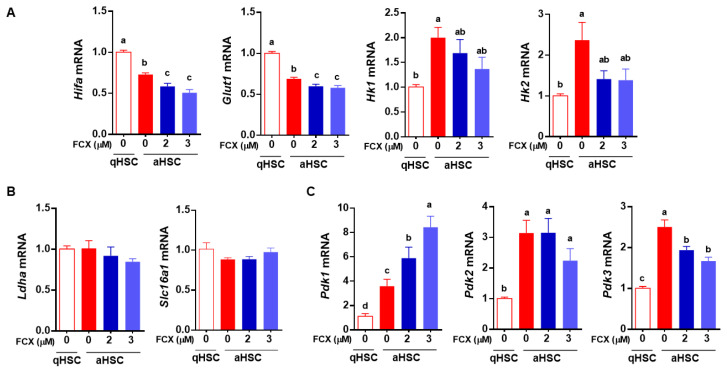
Effect of FCX on the expression of genes involved in glucose metabolism in aHSC. Primary mouse HSC were isolated and cultured for 1 day or 7 days to represent qHSC or aHSC, respectively. FCX (2 or 3 µM) was treated during HSC activation. (**A**) Expression of genes involved in glycolysis, including *Hif1a*, *Glut1*, *Hk1*, and *Hk2*. (**B**) Expression of genes involved in pyruvate metabolism, including *Ldha* and *Slc16a1*. (**C**) Expression of genes involved in pyruvate dehydrogenase activity, including *Pdk1*, *Pdk2*, and *Pdk3*. Bars not sharing a common letter are significantly different (*p* < 0.05). Mean ± SEM.

**Figure 4 nutrients-14-01902-f004:**
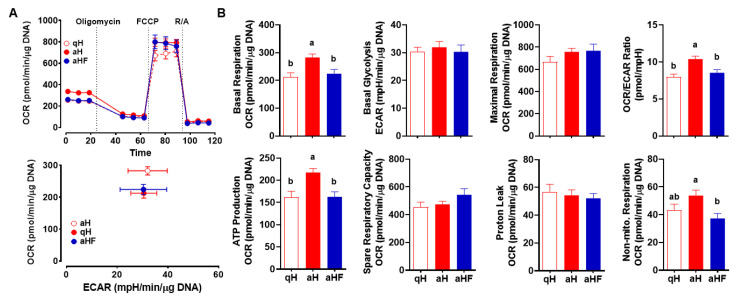
Inhibitory action of FCX in the increase in mitochondrial respiration during HSC activation. Primary HSC isolated from mice were cultured for 1 day or 7 days in the absence or presence of 3 µM FCX to represent qHSC (qH), aHSC (aH), or FCX-treated aHSC (aHF), respectively. MitoStress test (**A**,**B**) was conducted. Bars not sharing a common letter are significantly different (*p* < 0.05). Mean ± SEM.

**Figure 5 nutrients-14-01902-f005:**
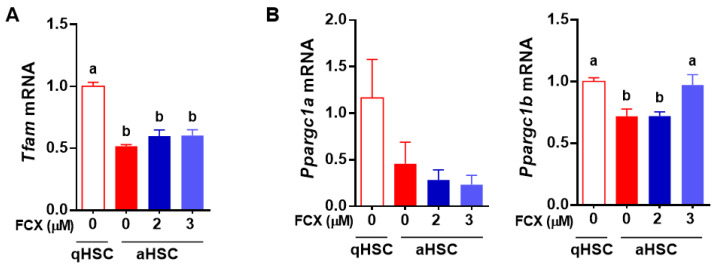
Effect of FCX on the expression of genes involved in mitochondrial respiration in aHSC. Primary mouse HSC were isolated and cultured for 1 day or 7 days to represent qHSC or aHSC, respectively. FCX (2 or 3 µM) was treated during HSC activation. Expression of *Tfam* (**A**) and *Ppargc1a* and *Ppargc1b* (**B**) in qHSC and aHSC treated with or without FCX. Bars not sharing a common letter are significantly different (*p* < 0.05). Mean ± SEM.

## Data Availability

Not applicable.

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
