# Peer review of "Fucoxanthin Attenuates the Reprogramming of Energy Metabolism during the Activation of Hepatic Stellate Cells"

_nutrients, 2022, doi:10.3390/nu14091902_

Round 1

Reviewer 1 Report

Here, the authors investigated the effects of fucoxanthin on HSC-specific energy metabolism. The findings are quite straightfoward regarding the question(s) the authors set out to answer. Nevertheless, I do have a few minor comments/suggestions that may improve the quality of this paper. 

  1. Do the authors have data on the morphology of the mitochondria - cristae structures in HSCs?
  2. What is the proposed mechanism for these observations?
  3. Authors should consider including data on the effects of fucoxanthin in vivo - using fibrosis model. 

Author Response

Here, the authors investigated the effects of fucoxanthin on HSC-specific energy metabolism. The findings are quite straightfoward regarding the question(s) the authors set out to answer. Nevertheless, I do have a few minor comments/suggestions that may improve the quality of this paper. 

  1. Do the authors have data on the morphology of the mitochondria - cristae structures in HSCs?

This is a great point. However, the primary goal of this study was to evaluate the effect of fucoxanthin on cellular energetics, focusing on glycolysis and mitochondrial respiration. Therefore, investigation of mitochondrial structure is beyond the scope of this study.

  1. What is the proposed mechanism for these observations?

The present study is novel in that it is first study demonstrating the role of fucoxanthin in modulating energy phenotype of cells. Although it is not clear how fucoxanthin elicits the changes in metabolic phenotypes of hepatic stellate cells, it may be related to its effect on mitochondrial structure as we reported before in our study on astaxanthin (M. Bae, Y. Lee, Y-.K. Park, D. Shin, P. Joshi, S. Hong, N. Alder, Sung I. Koo, J-.Y. Lee. Astaxanthin attenuates the increase in mitochondrial respiration during the activation of hepatic stellate cells. J Nutr Biochem 2019; 71:82-89). Alternatively, antioxidant effects of fucoxanthin may contribute to our findings. Future studies will be conducted to dissect molecular mechanisms of action.

  1. Authors should consider including data on the effects of fucoxanthin in vivo - using fibrosis model. 

This is a great suggestion. We have conducted an in vivo study and another manuscript is under review.

Reviewer 2 Report

  1. At the time of this discussion, there are reports that it showed antioxidant capacity, but it is not shown what kind of active oxygen was removed. Generally, there are four types of active oxygen in a broad sense, but hydrogen peroxide and hydroxyl radical are considered to be harmful to living organisms, so I would like to know what kind of results have been obtained in this field.
  2. It would be even better to mention the existence of AMPK in the mitochondrial discussion.

Author Response

  1. At the time of this discussion, there are reports that it showed antioxidant capacity, but it is not shown what kind of active oxygen was removed. Generally, there are four types of active oxygen in a broad sense, but hydrogen peroxide and hydroxyl radical are considered to be harmful to living organisms, so I would like to know what kind of results have been obtained in this field.

In our previous publication, we measured the effect of fucoxanthin on cellular ROS accumulation in hepatic stellate cells using dichlorofluorescein and demonstrated foucoxanthin inhibited TGFb1-induced ROS accumulation (M-.B. Kim, M. Bae, S. Hu, H. Kang, Y-.K. Park, J-.Y. Lee. Fucoxanthin exerts anti-fibrogenic effects in hepatic stellate cells. Biochem Biophy Res Comm 2019; 513 (3): 657-662. doi: 10.1016/j.bbrc.2019.04.052). As the focus of the current study is not antioxidant effects of fucoxanthin, we did not dissect types of ROS.

  1. It would be even better to mention the existence of AMPK in the mitochondrial discussion.

This is an interesting point. However, we have not measured any parameters related to AMPK and there was no strong indication that AMPK might play a role in exerting the effect of fucoxanthin. Therefore, we believe it is better not to include AMPK in our discussion. We hope the reviewer understands our stance.